# Characterization of the Sarcidano Horse Coat Color Genes

**DOI:** 10.3390/ani12192677

**Published:** 2022-10-05

**Authors:** Giovanni Cosso, Vincenzo Carcangiu, Sebastiano Luridiana, Stefania Fiori, Nicolò Columbano, Gerolamo Masala, Giovanni Mario Careddu, Eraldo Sanna Passino, Maria Consuelo Mura

**Affiliations:** Department of Veterinary Medicine, University of Sassari, Via Vienna 2, 07100 Sassari, Italy

**Keywords:** animal genetic resources, coat color, Sarcidano Horse, local genotypes

## Abstract

**Simple Summary:**

Currently, considerable attention is paid to the conservation of biological diversity, with the aim of saving low-spread populations from extinction. This study aimed to contribute to the general knowledge of the Sarcidano Horse, a small population autochthonous of the Sardinia island. Due to their semi-feral condition, it is assumed that there have been no crossings with domestic breeds. Since it is well known that the variety of coat colors in many current breeds derives from domestication and human selection, a first step towards a deeper knowledge of this breed has focussed on the study of the genetic basis of the coat colors. DNA from 70 Sarcidano horses has been analyzed to characterize *MC1R* and *ASIP* genes, responsible for the three basic coat colors: bay, black and chestnut. The results showed a clear prevalence of the chestnut color, a limited number of black and a very small presence of bay coats. Moreover, Sarcidano Horse showed only the basic coat color set, with no color dilution or spotting, which suggests the lack of crossbreeding with other domestic breeds and the state of genetic isolation of this population. Efforts in maintaining healthy this ancient genetic resource, through accurate genetic and phenotypic controls, look of vital importance.

**Abstract:**

The goal of this study was to contribute to the general knowledge of the Sarcidano Horse, both by the identification of the genetic basis of the coat color and by updating the exact locations of the genotyping sites, based on the current EquCab3.0 genome assembly version. One-hundred Sarcidano Horses, living in semi-feral condition, have been captured to perform health and biometric checks. From that total number, 70 individual samples of whole blood were used for DNA extraction, aimed to characterize the genetic basis of the coat color. By genotyping and sequencing analyses of the *MC1R* Exon 1 and *ASIP* Exon 3, a real image of the coat color distribution in the studied population has been obtained. Chestnut and Black resulted in the most representative coat colors both from a phenotypic and genotypic point of view, that is suggestive of no human domestication or crossbreeding with domestic breed. Due to its ancient origin and genetic isolation, an active regional plan for the conservation of this breed would be desirable, focused on maintenance of resident genotypes and genetic resources. Collection and management of DNA, sperm, embryos, with the involvement of research centers and Universities, could be a valid enhancing strategy.

## 1. Introduction

The Sarcidano Horse is a very small population of semi-feral horses, autochthonous of Sardinia, the second largest island in the Mediterranean Sea. The origins of these animals are uncertain; it is assumed their descendance from ancient Iberian horses introduced in Sardinia during the Aragonese domination [1]. Sarcidano Horse frequently has supernumerary bilateral upper premolars, which have disappeared in modern horse breeds, placing this breed on an archaic model (similarly to the Przewalski’s Horse). Previous research confirmed the marginal impact of crossbreeding on the Sarcidano Horse mitochondrial gene pools and the lack of recent gene flow from outside [2], suggesting that this breed represents an important resource to be preserved, but also an important model to be studied.

Due to the ancient origin and to the geographic isolation this breed should have a conservation program focused on maintaining the typical phenotype features and avoiding crossbreeding. Unfortunately, there are no real conservation plans for this breed at the moment. Among the major criticalities, obviously the first aspect to consider in native breeds is the low number within the population, which imposes the need for adequate mating strategies, the application of the most modern genetic improvement techniques to limit inbreeding and, above all, technical and economic assistance to the few breeders who take care of these animals so that they can adhere to the Stud Book, mating plans and functional checks.

Since it is well known that the great variety of coat colors present in many current breeds derives from domestication and human selection [3], a first step towards a deeper knowledge of this breed could be to study the genetic basis of the coat colors. According to the breed Standard, the Sarcidano Horse may be chestnut, black, bay, and, more rarely, grey. Although the coat color of an individual is visible, its real detection based exclusively on phenotype is often difficult. Two genes are primarily responsible for determination of base coat color in the horse: melanocortin-1-receptor (*MC1R*), encoded by the Extension (*E*) locus, and its peptide antagonist agouti-signaling-protein (*ASIP*), encoded by the Agouti (*A*) locus. These two loci are linked by a close epistasis relationship, where *MC1R* is epistatic to *ASIP*, so controlling the relative amounts of melanin pigments in mammals [4]. More precisely, the *MC1R* dominant allele E^E^ (hereafter simply referred as E) determines the production of black pigment (eumelanin), while the recessive E^e^ (=e) allele determines the production of a red-yellow pigment (pheomelanin) [5,6]. The “e” allele is produced by a C > T mutation [6] within the *MC1R* gene Exon 1 (chr:3: 36,979,560). Conversely, in the *ASIP* locus, the dominant A^A^ (A) allele encodes production of the agouti signaling protein, which has the ability to block the melanocortin receptor 1 function in the melanocytes. When this receptor is blocked, there is no stimulus for the production of eumelanin, and only pheomelanin synthesis occurs [7]. A recessive 11-bp deletion in *ASIP* Exon 3, causing the “a” allele, produces a frameshift that causes a loss of function in the agouti signaling protein, resulting in the uniform production of eumelanin [5]. Combination of specific genotypes at *MC1R* and *ASIP* loci result in the three basic phenotypes of horse coat colors: black, bay and chestnut [8].

This study aimed to contribute to increase the knowledge about the Sarcidano Horse, by identifying the genetic basis of the coat color, also establishing how the colors are genetically distributed within the existing population, and by updating the exact locations of the genotyping sites, based on the current EquCab3.0 genome assembly version.

## 2. Materials and Methods

### 2.1. Ethics Statement

All experimental procedures were reviewed and approved by the Organism in charge for the Animal Welfare and Experimentation (OPBSA) of the University of Sassari (Protocol number: 2018UNISSMEV 0000177).

### 2.2. Animals

Due to insularity and the harsh conditions of marginal areas in which it lives, Sarcidano Horse features a small size, with a wither’s height range 115–145 cm (mean 130 ± 15 cm). In addition to being smaller than the other Italian horse breed, the Sarcidano horse has distinctive phenotypic features. The head is rather heavy and roughly attached, the profile straight, the ears mobile and the eyes and nostrils large. The neck is muscular, with thick mane. The shoulder is fairly straight, the withers high and the croup short but muscular. The legs are short, strong and well conformed, with broad and solid joints. Since 2003, the Sarcidano Horse has been recognized as an independent breed [9,10], and it is presently recorded in the Stud Book, managed by the national association for the small local donkey and horse Italian breeds, since 2018 (ANAREAI), as a result of the Legislative decree no. 52 of the year 2018 [11,12]. Groups mainly consist of solitary subjects (without herd); herds with a dominant stallion, about 8–10 mares and young foals; herds of only sexually mature young males. The last census retrieved from FAO (last update 28 December 2021) recognized 118 total subjects, with only 3 stallions and 24 mares currently registered in the Stud Book [13]. Some differences were found checking these data on the ANAREAI website [11], in which resulted 109 registered Sarcidano Horses. However, both these values have to be considered underestimated because they reflect only subjects to whom it was possible to insert the microchip (transponder), and that have actually been registered. Furthermore, many microchipped subjects are currently unregistered, which demonstrates the small general interest around this breed and makes it more difficult to establish the real number of the living horses. The actual estimated adult male/female ratio is 1/5–6. This situation, although favorable in terms of variability and possibility of mating, nevertheless predisposes males to excessive competition and fights for dominance on mares’ herds. In order to prevent these fights from causing injuries, traumas and even the death of the animals involved, some males have been neutered based on their physical and health conditions. This action was also necessary for the creation of stable groups consisting of a male and some females.

### 2.3. Experimental Design

After a severe bushfire it was necessary to identify, to assess the sanitary status and to schedule the relocation of the Sarcidano horse population to other more suitable areas. For these reasons the animals were anaesthetized intramuscularly by a single-dart injection with a combination of tiletamine/zolazepam/detomidina/acepromazine mixed in the same syringe in small capture paddocks set up in different areas. Thus, a total of 100 Sarcidano horses, living in semi-feral condition in the Laconi municipality (province of Oristano), has been captured among February and June of the year 2016, also for microchip implantation, as a permanent form of individual identification and registration in the Stud Book. During anesthesia, the horses were subjected to various checks: biometric measures (height at the withers; length of the trunk; thoracic circumference; length and width of the head; circumference of the shin); health checks (stool collection for endoparasite control, sampling and recognition of ectoparasites and evaluation of infestation extent; external control for eventual superficial lesions).

### 2.4. Blood Sampling and Genotyping

From the captured 100 Sarcidano horses, 70 randomly chosen were intended for our laboratory for genetic controls. Total DNA was extracted from individual 200 μL of peripheral whole blood samples, by automated extraction using the NucleoSpin^®^ Blood kit (Macherey-Nagel, Düren, Germany), following the provided protocol. Polymerase chain reaction (PCR) technique was carried out to amplify parts of *MC1R* and *ASIP* genes as they are the two strongest candidate genes that influence coat color in horses.

#### 2.4.1. Amplification

An amount of 150 ng of the obtained genomic DNA was subjected to PCR, using two pairs of primers, designed based on the latest horse genome version EquCab3.0 within the *MC1R* and *ASIP* genes sequence. More precisely, the couple of primers Forward: 5′–CCT CGG GCT GAC CAC CAA CCA GAC GGG GCC–3′ and Reverse: 5′–CCA TGG AGC CGC AGA TGA GCA CAT-3′ delimited a 317-base pair (bp) fragment, corresponding to part of the unique Exon 1 of the *MC1R* gene; while primers Forward: 5′–CTT TTG TCT CTC TTT GAA GCA TTG–3′ and Reverse: 5′– GAG AAG TCC AAG GCC TAC CTT G–3′ delimited a 102/91-bp polymorphic fragment, corresponding to the entire Exon 3 of the *ASIP* gene.

PCR reaction for both loci was carried out in 25µL final volume, containing 1X PCR Buffer (minus MgCl_2_) (20 mM Tris-HCl (pH 8.0), 40 mM NaCl, 2 mM Sodium Phosphate, 0.1 mM EDTA, 1 mM DTT, stabilizers, 50% (*v*/*v*) glycerol); 2.0 mM of MgCl_2_; 200 μM of each dNTPs, 0.2 μM of each primer, and 0.25 Units (U) of Taq DNA polymerase (HOT FIREPol^®^ Polymerase, Solis BioDyne, Tartu, Estonia) and ultrapure water DNase/RNase free (Water PCR grade, Solis BioDyne, Tartu, Estonia) up to 25 μL. Thermocycling on a MAXYGENE II machine (Axygen^®^ Tewksbury, MA, USA) was carried out for 35 or 40 cycles in total for *MC1R* and for *ASIP* amplicons. PCR conditions were as follows: activation of the *Taq* DNA polymerase at 95 °C for 15 min; initial denaturation at 95 °C for 5 min, followed by 35 (for *MC1R* amplicon) or 40 (for *ASIP* amplicon) cycles of denaturation at 95 °C for 30 s, annealing at 58 °C or 60 °C for *MC1R* and *ASIP*, respectively, for 30 s, elongation at 72 °C for 30 s; final extension at 72 °C for 10 min.

Ten microliters of the obtained amplicons products were electrophoresed at at steady voltage of 110 V for 30 min, in a 2% ultrapure agarose gel (*w*/*v*) (iNtrRon Biotechnology, Sangdaewon-Dong, Korea) added with 9 μL of RedSafe stain (iNtrRon Biotechnology, Sangdaewon-Dong, Korea), in 1X TAE electrophoresis buffer, and visualized by ultraviolet transillumination (UVItec, Cambridge, UK), together with a 100-bp Ladder (GeneRuler, Thermo Scientific™, Waltham, MA, USA).

#### 2.4.2. Digestion

Amplicons from *MC1R* Exon 1 were then digested in a total volume of 30 μL containing 10 μL of the PCR amplification reaction, 1X rCutSmart Buffer (50 mM Potassium Acetate, 20 mM Tris-Acetate, 10 mM Magnesium Acetate, 100 μg/mL Recombinant Albumin, pH 7.9 at 25 °C), 5 U of *TaqI* endonuclease, by following the manufacturer’s instruction (New England BioLabs, Beverly, MA, USA), and nuclease-free water up to final volume. The digestion mix were incubated in thermostatic bath at 65 °C for 15 min (Time-Saver™ Qualified Restriction Enzymes). The resulting fragments were resolved by electrophoresis at 80 V for 90 min in a 2% ultrapure agarose gel (*w*/*v*) (iNtrRon Biotechnology, Sangdaewon-Dong, Korea) added with 9 μL of RedSafe stain (iNtrRon Biotechnology, Sangdaewon-Dong, Korea), in 1X TAE electrophoresis buffer, and visualised by ultraviolet transillumination (UVItec, Cambridge, UK), in parallel with a 100 bp Ladder (GeneRuler, Thermo Scientific™, Waltham, MA, USA).

#### 2.4.3. Genotyping and Sequencing

Genotyping at the *MC1R* locus, was carried out digesting the obtained amplicons with the *TaqI* endonuclease. This restriction enzyme cut site recognizes a C > T substitution, producing a single nucleotide polymorphism (SNP) at position 180 of the amplified fragment. By this way *TaqI* endonuclease can recognize three available genotypes designated as T/T, T/C and C/C.

Conversely genotyping at the *ASIP* locus was obtained directly by PCR, based on a 11-bp deletion that produced different fragment size after electrophoresis run. This condition allows recognition of three possible genotypes designated A/A, A/a and a/a within the *ASIP* gene Exon 3.

Genotypes from the *MC1R* and *ASIP* loci and their combination determined the base coat color of the sampled horses as specified in the Results section.

Two samples from each genotype of both the *MC1R* and *ASIP* genes were then sequenced in forward and reverse direction by a commercial service, in order to confirm genotyping and to compare the old sequence with the current one.

#### 2.4.4. Update on Polymorphisms Position within the Current Horse Genome Version

The search for the correspondences between the previously recognized polymorphic sites within the latest horse genome assembly was carried out by comparison of existing results on *MC1R* and *ASIP* loci polymorphisms with those obtained by the present research, referred to the latest EquCab3.0 genome assembly (GCF_002863925.1). Accessible information about number and composition of the exons composing the two analyzed genes, has been rationalized and reorganized through the comparison and finding evidence of reciprocity. The currently available information about the exact position of polymorphic sites responsible for the different genotypes has been updated here, allowing a more complete interpretation of the results.

#### 2.4.5. Statistical Analysis

Due to the small number of the tested horses, allelic frequencies were determined by direct counting of the observed genotypes.

To reveal the potential association between genotypes of *MC1R* and *ASIP* and phenotypes of the horse coat colors, a χ^2^ test for independence was performed using R statistical software (Version 4.1.2 R Core Team 2021 R: A language and environment for statistical computing. R Foundation for Statistical Computing, Vienna, Austria; https://www.R-project.org/, accessed on 1 July 2022.

## 3. Results

Captured horses resulted in 63 females and 37 males, aged from 15 days to 17 years old. Age determination, by teeth observation, exhibited the following distribution: 51 were young (aged 0–4 years); 21 were adults (5–10 years) and 20 were over 10 years old (even up to 17 years). The 70 individuals here genotyped were 27 males and 43 females, distributed in 37 young, 16 adults and 17 elderly, following the same age classification explained above.

All the 70 samples processed have been successfully amplified and genotyped both at the *MC1R* and *ASIP* locus. Individual information about *ASIP* and *MC1R* genes genotype, age, sex and phenotypic/genetic coat colors in the studied population are available in Appendix A.

The sequencing result did not produce a useful reading for the *ASIP* gene fragment, due to its shortness, so that all the genotypes identified here were obtained as follows.

PCR products containing the mutation positions in *MC1R* (317-bp)—later digested by *TaqI* endonuclease—and in *ASIP* (102-bp) genes, allowed identification of the genotypes set (E/E, E/e, e/e and A/A, A/a, aa/) for *MC1R* and *ASIP* locus, respectively (Figure 1). A summary of the *MC1R* and *ASIP* genes genotypes, and allele and genotype frequencies for the horse coat colors are shown in Table 1, Table 2 and Table 3.

The comparison between previous polymorphic sites positions and the latest EquCab3.0 horse genome assembly version resulted an updated set of information about these loci, described below in detail.

### 3.1. MC1R Locus Genotypes

The *MC1R* gene is on chromosome 3 in horses, and it extends from the 36,979,313 to 36,980,266 position of the latest horse genome version EquCab3.0 (GCF_002863925.1). The entire *MC1R* gene consists of only one exon of 954 base pair (bp) in length. A single fragment of 317-bp in length was obtained by PCR analysis of the *MC1R* locus, corresponding to part of the unique Exon 1.

After restriction fragment length polymorphism (RFLP) analysis, carried out through *TaqI* restriction enzyme digestion, one polymorphic site was found at position 180 of the amplified fragment, corresponding to chr3: 36,979,560 of the EquCab3.0 genome assembly, involving a C with a T substitution. This polymorphic site corresponded to the *MC1R* C901T mutation recognized in previous studies [6,14]. This SNP produces three different genotypes, here named E/E, E/e and e/e, and recognized, after electrophoretic run, on the basis of the fragments number and size. More precisely, a unique 317-bp fragment corresponded to C/C (or E/E) dominant homozygous genotype; a 180 + 137-bp electrophoretic pattern matched with T/T (e/e) recessive homozygous genotype, and finally a 317 + 180 + 137 electrophoretic pattern corresponded to C/T (E/e) heterozygous genotype (Figure 1). The *MC1R* genotype distribution exhibited 47 horses carrying recessive homozygous e/e (T/T) genotype, corresponding to 67%, 20 horses carrying heterozygous E/e (C/T) genotype, corresponding to 29%, and only three horses carrying dominant E/E (C/C) genotype, corresponding to 4% of the analyzed population (Table 2). Allele frequency was 0.19 for the “E” allele and 0.81 for the “e” allele (Table 2).

### 3.2. ASIP Locus Genotypes

The ASIP gene is on chromosome 22 in the horse and ranges from the nucleotide 26,009,341 to 26,072,655 of the EquCab3.0 horse genome assembly, spanning 63,315 nucleotides. It consists of four exons in the principal isoform and of five putative exons in the X1 isoform. Before the EquCab3.0 sequence was available, the mutation responsible for the phenotypic coat color change was believed to occur in Exon 2 [15], while it is now known that this mutation falls in Exon 3.

The Exon 1 counts 499-bp in length, ranging from 26,009,341 to 26,009,839 position; it is the largest of the 4 exons composing the entire *ASIP* gene. The Exon 2 covers 170-bp, from 26,065,977 to 26,066,136 position; Exons 1 and 2 are separated by an intron of approximately 56.1-kbp. The Exon 3 is the shortest, ranging from 26,067,437 to 26,067,501, but within its 65-bp sequence falls the causative mutation. It is separated from the Exon 2 by the Intron 2, which covers 1.3-kbp, and from the Exon 4 by the Intron 3, spanning approximately 2.3-kbp. Finally, the Exon 4 covers 177-bp, from position 26,069,795 to 26,069,971.

Here we amplified a 102-bp fragment, corresponding to the entire Exon 3—consisting of only 65 nucleotides—and small parts of the adjacent Introns 1 and 2. This entire 102-bp fragment identified the dominant, wild type “A” allele, that inhibit the eumelanin production, while an 11-bp deletion mutation in the above fragment leads to the mutant, recessive “a” allele (consisting in the resulting 91-bp fragment after amplification) (Figure 1). This 11-bp deletion mutation consists in the elimination of the nucleotide sequence CAGAAAAGAAG from chr22: 26,067,476 to 26,067,486 position (rs396813234) of the EquCab3.0 genome assembly (GCF_002863925.1).

The *ASIP* genotype distribution was similar to those observed in *MC1R* locus. Indeed, 44 horses carried the recessive a/a homozygous genotype, corresponding to 63%, 24 horses carried heterozygous A/a genotype, corresponding to 34%, and only two horses carried dominant homozygous A/A genotype, corresponding to 3% of the studied population. Consequently, allele frequency was 0.20 for the “A” allele and 0.80 for the “a” allele (Table 3). In both the loci, mutant recessive allele resulted more frequently than the wild, dominant type.

### 3.3. Colour Coat Assessment

The most representative color coat in Sarcidano Horse was the chestnut, both from a genetic and phenotypic point of view: a number of 47 horses of the analyzed 70 were Chestnut, corresponding to 67% of the studied population. The 27% were Black (19 horses) and only 6% (4 horses) were Bay. Some differences were observed in the color shade of the chestnut and bay coat, from light to dark, depending on the allele combination. The most frequent allele combinations resulted in e/e-a/a that correspond to cherry-liver chestnut phenotype, which is the most visible coat color in the Sarcidano population. Another lighter chestnut phenotype was present, although in only two subjects, obtained by the e/e-A/A allele combination. Bay coat color resulted both phenotypically and genetically poorly represented in the Sarcidano Horse, with 4 subjects carrying only E/e-A/a allele combination, and none carrying one of the other possible three genotypes for bay color (E/e-A/A; E/E-A/A and E/E-A/a). The χ^2^ test results showed that the phenotypes of horse coat colors were significantly related with the genotypes and alleles of *MC1R* and *ASIP* (*p* < 0.001, Table 1 and Table 2).

Finally, Black color coat resulted moderately represented in the studied population with 16 subjects carrying E/e-a/a genotype and 3 carrying E/E-a/a genotype. In total 8 horses (2 males and 6 mares) were phenotypically Grey, but their genetic color resulted in all cases Chestnut, i.e., 4 carrying A/a-e/e and 4 carrying a/a-e/e genotype.

## 4. Discussion

In genetics, the knowledge of the link between genotype and phenotype is crucial to understand how phenotypic variation influences strength and timing of selection through evolutionary change within a population.

Coat, hair and skin colors depend on the pigment produced in the melanocytes. In horses, the base coat color is mainly influenced by two candidate genes: melanocortin -1 receptor (*MC1R*) and agouti signaling protein (*ASIP*), that work together to control the coat color phenotypic trait [7,16,17].

The *MC1R* gene produces a protein called melanocortin 1 receptor (MC1R), which plays an important role in normal pigmentation, and it is primarily located on the surface of melanocytes (specialized cells that produce a pigment called melanin). Melanin is the substance that gives color to skin, hair, and eyes, and it is also found in the retina, where it plays a role in normal vision [18].

When the MC1R is activated, by the melanocyte stimulating hormone (MSH) produced by the intermediate pituitary, stimulates melanocytes to make eumelanin; while if the receptor is not activated or if it is blocked, melanocytes produce pheomelanin instead of eumelanin [19].

Common variations (polymorphisms) in the *MC1R* gene result in differences in skin and coat color, although also other genes contribute to the normal mammalian pigmentation. These *MC1R* polymorphisms prevent eumelanin production by melanocortin 1 receptor, inducing output of pheomelanin.

The melanocortin 1 receptor plays also an important role in the body’s immune and inflammatory responses. The receptor’s function in these cells is still poorly known. *MC1R* gene is also known as *E locus* (where E means *Extension*), and two allele are recognized within this locus: the dominant “E” allele, determining the production of eumelanin; and the recessive “e” allele determining the production of pheomelanin [20]. A SNP located at position chr:3: 36,979,560 of the EquCab3.0 genome assembly, within *MC1R* gene nucleotide sequence causes a T to a C transition, that leads an amino acid substitution from Serine (TCC) to Phenylalanine (TTC) at position 83 of the protein chain (NCBI Reference Sequence: NP_001108006.1). This mutation is involved in a transmembrane domain of the transport protein and its substitution is likely to disrupt the secondary local structure. The normal allele, with C was named “E” and the recessive one, with C replaced by T, “e” [21]. This recessive allele generated by the above mutation leads to the production of only pheomelanin within melanocytes [6]. Thus, pheomelanin production depends on melanocytes that are homozygous for the recessive e/e genotype at *MC1R* locus which therefore cannot be activated by the MSH [6,22]. When the melanocortin -1 receptor is activated by the MSH, it triggers a series of chemical reactions within the melanocytes, leading to production of eumelanin. In fact, melanocytes are capable of producing both eumelanin and pheomelanin, but when the animal carry the e/e genotype, the receptor is defective, making it unable to properly transmit the information passed by MSH, thus leading to production of pheomelanin, only [6,22]. The net result of the forms of this gene is therefore an animal with a black coat when there is at least one dominant allele (E/E or E/e), or a Chestnut horse (reddish color) when it is homozygous to the recessive allele (e/e). The recessive allele (e) is a mutation of the wild form (E), which occurred about 7000 years BP [23].

The other gene involved in the skin and coat color in mammals is the *ASIP* gene: the dominant “A” allele encodes production of a protein called agouti signaling protein, which has the ability to block the melanocortin -1 receptor in existing melanocytes in the body of the horse, but not in the extreme points (mane, tail, ear edges, and lower legs). Therefore, an eumelanic horse (E/-) with a dominant allele at the Agouti locus (A/-) yields a Bay horse—a horse with a reddish to brownish body color and black points. When the receptor is blocked, there is no stimulus for the production of eumelanin (as explained above), and only pheomelanin synthesis occurs [6].

About 8000 years BP, a recessive mutation occurred in the ASIP locus, yielding black color throughout the entire body of the horse [23]. For this to happen, however, two recessive a/a allele are required. Interestingly, the agouti signaling protein is effective only in eumelanin-producer horses (i.e., carrying E/E or E/e genotype), whereas, in chestnuts (whose genotype is e/e) melanocortin-1 receptor is defective, consequently the protein has no effect. Similar to in other domesticated species, also in horses, artificial selection started from domestication process, caused mutation in color associated genes, followed by their fixation and increase in frequency [3]. In the history of domestic horses, crosses among breeds/lineages were common, resulting in a widespread distribution of coat-color-associated alleles, starting with their initial introduction in the gene pool of domestic horses. Therefore, artificial selection is the main factor responsible for the large phenotypic variation observed in domesticated animals today [3]. Through time, humans have actively encouraged coat color changes and consequently the proliferation of new coat color alleles, often fascinated by the new combinations thus produced [24]. The alleles for the basic colors (bay, black, chestnut) in the *MC1R* gene [14] and in the ASIP gene [5] are at least 6300 years old and were already found in pre-domestic times [25,26,27]. Consequently, the color alleles for bay, black, and chestnut occur in nearly all breeds, but their proportion can differ on the base of the breed specific history and geographical location; for example, in Misaki horses, a small local breed in southwestern Japan the most representative coat color is Black, with very small number of Chestnut horses [28], in contrast with what was found here. The absence of coat color dilutions or spotting in Sarcidano Horse, allows to confirm the state of genetic isolation in which this population has evolved and has reached the present day.

One of the most interesting aspects of the present study is that the attribution of the coat color in horses, based on visual observation alone, is not always reliable. Additional difficulties on identification arise when considering several situations, some of which are explained below.

Different nomenclatures for the same coat color are not rare, depending on different countries, or regions within the same country. Moreover, different classifications in horse coat color derive from specific breed standard: although genetically the black color is possible in the horse as in all other mammals, some breed standards do not admit it among the coat colors and identify all dark coats as Dark Bay.

A subjective perception of a particular shade can create a difficult identification of horse’s color; thus, only genetic characterization is correct and reliable. The influences of external and individual factors, such as age, time of year, living condition, and nutritional status can make it difficult to identify a horse’s color. Thus, a Black horse kept always outdoors, could exhibit a “fading” effect on coat color, making it easily confounded with a Dark Brown horse. Conversely, the Dark Bay horse coat color is a black mane, tail and legs, and very dark brown or reddish hair at the head, neck, back and hip, which makes it confusing with a Black coat faded by sunlight.

Horses with brown coat color have black pigment in the mane, tail and legs, and have a nearly black phenotype of their whole body, but reddish or tan pigments around their eyes, muzzle, and abdomen between the elbow and stifle. This makes them easily confused with Liver-Chestnut coat color. At the same time, Chestnut horses are reddish with no black color on their entire body including the mane, but the shade of red can be so intense, that it can be confused with a Bay-Brown coat color.

Grey horses are born with an original coat color (such as black, bay, or chestnut) but gradually lose hair pigmentation at ages (6 to 8 years) but maintain the dark skin [29]. The Grey coat color is due to the presence of dominant allele—G—at the grey locus. The genotype of grey horses will be either G/G or G/g and the horse without grey gene is symbolized as g/g. Grey is epistatic to all coat color genes except white and grey horses must have at least one grey parent [30]. Thus, defining a horse as Grey tells us nothing about its real color and, very importantly, what coat color its offspring will inherit.

## 5. Conclusions

To conclude, this study produces a general contribution to define coat colors on a molecular level, in a population of semi-feral horses, autochthonous of Sardinia. Through domestication and human selection, the different horses’ breeds have acquired various coat colors, starting from the three recognized as basic: Chestnut, Bay and Black. Thus, the absence of spotting or color dilution in a breed suggests low phenotypic variability and could be indicative of lack of crossbreeding with other domestic breeds.

This study showed a high prevalence of the Chestnut coat color in the Sarcidano Horse, both from a phenotypic and genetic point of view. Moreover, Sarcidano Horse showed only the basic coat color set, which suggests a low impact of human selection and removes the probability of crossing with other domestic breeds in the past. In order to safeguard this ancient genetic resource, it seems vitally important to carry out more accurate genetic and phenotypic controls. It would be desirable to create an active regional plan for the conservation of this breed aimed to include it in an economic circuit able to enhance both this breed and the link with its territory of origin. Collection and management of DNA, sperm, embryos, with the involvement of research centers and Universities, could be a valid enhancing strategy.

## Figures and Tables

**Figure 1 animals-12-02677-f001:**
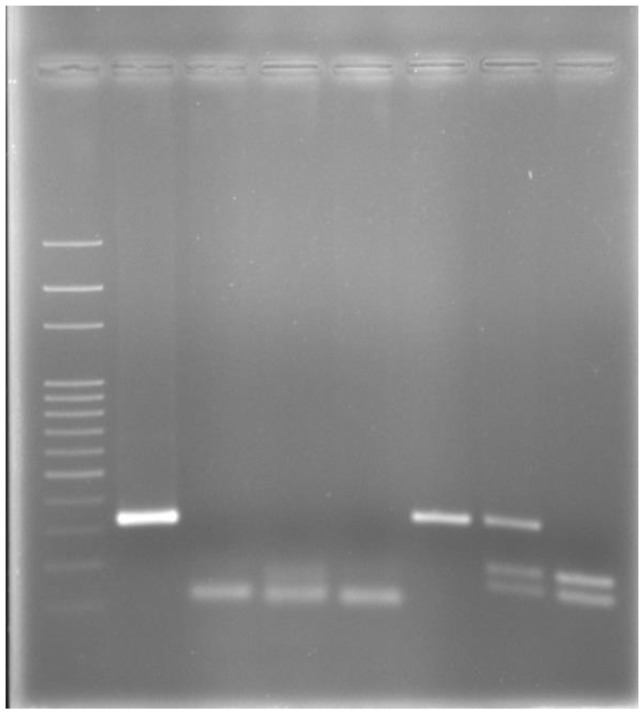
Electrophoresis after *MC1R* and *ASIP* amplification/genotyping in a 2% agarose gel: lane 1: 100-bp Ladder; lane 2: amplified *MC1R* Exon-1 fragment of 317-bp in length; lane 3: A/A homozygous dominant genotype without 11-bp deletion in any allele (102-bp band only); lane 4: A/a heterozygous genotype with 11-bp deletion in only one allele (102 + 91-bp bands); lane 5: a/a homozygous recessive genotype with 11-bp deletion in both the alleles (91-bp band only); lane 6: E/E or C/C dominant homozygous genotype (317-bp band only); lane 7: E/e or C/T heterozygous genotype (317 + 180 + 137-bp bands); lane 8: e/e or T/T recessive homozygous genotype (180 + 137-bp bands).

**Table 1 animals-12-02677-t001:** Genotypes and alleles distribution of the *MC1R* polymorphism in the coat color phenotypes of the 70 genotyped Sarcidano Horse.

		*MC1R* Genotype			*MC1R* Alleles	
		E/E	E/e	e/e	E	e
Coat color	Black	3	16	0	22	16
	Bay	0	4	0	4	4
	Chestnut	0	0	47	0	94

**Table 2 animals-12-02677-t002:** Genotypes and alleles distribution of the *ASIP* polymorphism in the coat color phenotypes of the 70 genotyped Sarcidano Horse.

		*ASIP* Genotype			*ASIP* Alleles	
		A/A	A/a	a/a	A	a
Coat color	Black	0	0	19	0	38
	Bay	0	4	0	4	4
	Chestnut	2	20	25	24	70

**Table 3 animals-12-02677-t003:** Combined distribution of *MC1R* and *ASIP* loci genotypes and base coat color phenotypes of the 70 genotyped Sarcidano Horse.

		*MC1R* Genotype		
		E/E	E/e	e/e
*ASIP* genotype	A/A	0	0	2 Chestnut
	A/a	0	4 Bay	20 Chestnut ^1^
	a/a	3 Black	16 Black	25 Chestnut ^1^

^1^ Four of which were phenotypically Grey.

## Data Availability

The data presented in this study are available on request from the corresponding author.

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
