# Peer review of "Characterization of the Sarcidano Horse Coat Color Genes"

_animals, 2022, doi:10.3390/ani12192677_

Round 1

Reviewer 1 Report

General comments

This manuscript deals with the genetic characterization of coat colour genes in Sardinia's small local horse breed. The rationale of the study is well justified and the possible implication of study in the horse breeds contest is sound in my opinion. The study is based on a not very big dataset, but I think it is sufficiently robust for the aim of the study, considering that it accounts for almost all the existent population of Sarcidano horses. All procedures adopted, data collection, sample processing, MC1R and ASIP genes sequencing, and statistical analysis are well described in the appropriate sections. The results and discussion are precise and reflect the main aim of the study. Additionally, the study on the MC1R and ASIP genes in this local small horse breed are novel to my knowledge. I do not have an English mother tongue, but I suggested a few tests aimed at improving the quality of the manuscript. I have no specific major concerns on this study, except on the description of the anagraphic register, for which I suggested specific changes where appropriate. Therefore, in my opinion, this study is almost worth to be published in the present form, with only some minor adjustments given below as specific points. These should be intended as changes to increase the quality of the text editing.

Specific comments

L 13. “…has focussed on the study….”.

L 40. “…in modern horse breeds, placing….”

L 44. “…to be preserved, but also an important model to be studied”.

L 52. “…adhere to the studbook”. The Legislative decree no. 52 of the year 2018 has suppressed all registers previously managed by the Italian Breeders’ Association (AIA) and has transferred all-new stud books for the small local horse populations to the ANAREAI, which is now in charge of the management of all studbooks belonging to these local breeds, including the Sarcidano horse breed. Please visit the site https://www.anareai.it/ to get a confirmation on the existence of this new breeders’ association.

L 76. “….to increase the knowledge…”.

L 94-95. Based on the previous comment on the new studbooks managed by ANAREAI, please modify accordingly.

L 97-99. Please check the data on the actual population of Sarcidano Horse on the ANAREAI website suggested above. Data are not totally in agreement with those reported on the website, although differences are very small.

L 120. “Anagraphic register” should be changed with Stud Book.

L 295. “The 27% were black (19 horses) and only 6% (4 horses) were bay.”.

L 301. Please delete the word “only”.

L 323. “… but their genetic allele combination resulted in all cases chestnut, i.e. 4 carryings….”

L 326-327. “In genetics, the knowledge of the link between genotype and phenotype is crucial to understand….”.

L 344. “…gives colour to skins, hair, and eyes.”.

L 350. Replace to make with to produce.

L 377 “…of pheomelanin, only”. Addition of a comma after the word pheomelanin.

L 389. “…the basic colours (bay,…..”.

L 414. “…. can create a difficult identification of horse’s colour.”

L 435. “…basic colours”

L 442. “This study produces….”

L 497-498. Please revise these 2 references based on the Legislative Decree no. 58 of 2018.

Author Response

Rev1

Comments and Suggestions for Authors

Rev1#point 1:   

General comments

This manuscript deals with the genetic characterization of coat colour genes in Sardinia's small local horse breed. The rationale of the study is well justified and the possible implication of study in the horse breeds contest is sound in my opinion. The study is based on a not very big dataset, but I think it is sufficiently robust for the aim of the study, considering that it accounts for almost all the existent population of Sarcidano horses. All procedures adopted, data collection, sample processing, MC1R and ASIP genes sequencing, and statistical analysis are well described in the appropriate sections. The results and discussion are precise and reflect the main aim of the study. Additionally, the study on the MC1R and ASIP genes in this local small horse breed are novel to my knowledge. I do not have an English mother tongue, but I suggested a few tests aimed at improving the quality of the manuscript. I have no specific major concerns on this study, except on the description of the anagraphic register, for which I suggested specific changes where appropriate. Therefore, in my opinion, this study is almost worth to be published in the present form, with only some minor adjustments given below as specific points. These should be intended as changes to increase the quality of the text editing.

Author’s response#point1: Thank you for your evaluable opinion. The correction made to the section dedicated to the Register and the description of the Stud Book has been greatly improved thanks to your suggestions and updates, and we are grateful. The other required changes have been made and are listed below.

Rev1#point 2:   

Specific comments

L 13. “…has focussed on the study….”.

Author’s response#point2: sentence changed in: “…a first step towards a deeper knowledge of this breed has focused on the study of the genetic basis of the coat colors.” (Lines 12-13)

Rev1#point 3:

L 40. “…in modern horse breeds, placing….”

Author’s response#point3: “in modern horses” changed in “in modern horse breeds”. (Lines 40-41)

Rev1#point 4:

L 44. “…to be preserved, but also an important model to be studied”.

Author’s response#point4: “but also an important model to be studied” added after “to be preserved” (Line 45-46)

Rev1#point 5:

L 52. “…adhere to the studbook”. The Legislative decree no. 52 of the year 2018 has suppressed all registers previously managed by the Italian Breeders’ Association (AIA) and has transferred all-new stud books for the small local horse populations to the ANAREAI, which is now in charge of the management of all studbooks belonging to these local breeds, including the Sarcidano horse breed. Please visit the site https://www.anareai.it/ to get a confirmation on the existence of this new breeders’ association.

Author’s response#point5: Many thanks for providing us with new information, we have changed all relative references accordingly.

Rev1#point 6:

L 76. “….to increase the knowledge…”. Enshrined

Author’s response#point6: “improving” changed in “to increase”. (Line 78)

Rev1#point 7:

L 94-95. Based on the previous comment on the new studbooks managed by ANAREAI, please modify accordingly.

Author’s response#point7: the sentence has been completely revised, according to your updates (Lines 98-100)

Rev1#point 8:

L 97-99. Please check the data on the actual population of Sarcidano Horse on the ANAREAI website suggested above. Data are not totally in agreement with those reported on the website, although differences are very small.

Author’s response#point8: thank you for your useful update. Actually, the number are quite different, but in both cases, we think they can be under-estimated values, as specified in the manuscript (Lines 111-116)

Rev1#point 9:

L 120. “Anagraphic register” should be changed with Stud Book.

Author’s response#point9: “Anagraphic Register” changed in “Stud Book” (Line 124)

Rev1#point 10:

L 295. “The 27% were black (19 horses) and only 6% (4 horses) were bay.”.

Author’s response#point10: “only 4 horses (6%) were bay.” changed in “…only 6% (4 horses) were bay.” (Line 305).

Rev1#point 11:

L 301. Please delete the word “only”.

Author’s response#point11: “only” deleted. (Line 311).

Rev1#point 12:

L 323. “… but their genetic allele combination resulted in all cases chestnut, i.e. 4 carryings….”

Author’s response#point12: “…but their genetic color in all cases 323 resulted in Chestnut (4 carrying A/a-e/e and 4 carrying a/a-e/e genotype).” Changed in “but their genetic color resulted in all cases Chestnut, i.e. 4 carrying A/a-e/e and 4 carrying a/a-e/e genotype”. (Line 336-337).

Rev1#point 13:

L 326-327. “In genetics, the knowledge of the link between genotype and phenotype is crucial to understand….”.

Author’s response#point13: The sentence “In the field of genetics, it is crucial to understand the link between genotype and phenotype of each individual, in order to understand how…” changed in “In genetics, the knowledge of the link between genotype and phenotype is crucial to understand how…” (Lines 339-340)

Rev1#point 14:

L 344. “…gives colour to skins, hair, and eyes.”

Author’s response#point14: The sentence “Melanin is the substance that gives skin, hair, and eyes their color.” Changed in “Melanin is the substance that gives color to skin, hair, and eyes.” (Line 357)

Rev1#point 15:

L 350. Replace to make with to produce.

Author’s response#point15: “make” changed in “produce”. (Line 371)

Rev1#point 16:

L 377 “…of pheomelanin, only”. Addition of a comma after the word pheomelanin.

Author’s response#point16: comma after “pheomelanin” added. (Line 398)

Rev1#point 17:

L 389. “…the basic colours (bay,…..”.

Author’s response#point17: “basic coloration” changed in “basic colors” (Line 419)

Rev1#point 18:

L 414. “…. can create a difficult identification of horse’s colour.”

Author’s response#point18: “…can alter the right coat color identification” changed in “…can create a difficult identification of horse’s colour.” (Lines434-435)

Rev1#point 19:

L 435. “…basic colours”

Author’s response#point19: “colorations” changed in “colors”. (Line 458).

Rev1#point 20:

L 442. “This study produces….”

Author’s response#point20: “makes” changed with “produces” (Line 465).

Rev1#point 21:

L 497-498. Please revise these 2 references based on the Legislative Decree no. 58 of 2018.

Author’s response#point21: The cited References have been properly changed.

Reviewer 2 Report

The manuscript reports the MC1R and ASIP genotypes determining coat color in 70 Sarcidano horses. The idea of this study is interesting, but the outcome demonstrated in the manuscript is quite disappointed. The manuscript contains numerous flaws and requires extensive editorial work and a major revision.

1) The Authors studied 70 horses of 100 Sarcidano horses. From their description, I understood that this horse population is highly inbred, and all these horses should be considered as the relatives. Strictly speaking, the Authors determined the frequencies of the alleles and genotypes in a large pedigree rather the frequencies in a population with randomly mating animals. In that case, the study has no sense without demonstrating the pedigree of these animals determined by horse managers or microsatellite profiling.

2) In situation when the pedigree information or the microsatellite profiling is unavailable, the Authors need at least to provide a supplement with the individual information for each animal including gender, age, phenotypic coat color, MC1R and ASIP alleles, MC1R and ASIP genotypes, the genetic coat color.

3) The restriction analysis was not confirmed by the DNA sequencing of the MC1R and ASIP alleles despite the Authors mentioned that method on page 4. All the alleles have to be confirmed and the fragments of the sequencing graphs with particular SNPs or a deletion must be included.

4) Figures 2 and 3 are unreadable and must be substantially improved. Do not include Figure 1 in the manuscript like you want to provide evidence that your PCRs work. Please, add one lane of the undigested PCR product for the comparison with all combinations of the allelic fragments in the digested PCR samples and the PCR fragments with a deletion polymorphism. Provide the better separation of the alleles on gels. It is easy to prepare new gels and take pictures of good quality.

5) Why did the Authors study just 70 horses? If 100 horses were available, it would be great to see the results for all animals. The Authors did not explain why 30 animals were neglected.

6) Discussion. Please, do not discuss the topics you did not analyze in your project. You did not study how both genes MC1R and ASIP work. There is no sense to provide the textbook description on this topic in Lines 333 – 389. Instead, please discuss the distribution of the MC1R and ASIP alleles in other horse breeds and wild/feral populations. Also, describe better the artificial selection of coat color in the process of horse domestication.

7) Many statements in the manuscript taken from other publications missed the references. For instance, the references are missed in Lines 55, 67, 261, 269, 346, 389.

9) Line 112: “After a severe bushfire…”. Please, provide more details about this bushfire and how it affected the Sarcidano horse population.

10) References. The Authors need to follow the journal’s requirements for the references. Please, compare the references 1 and 2 in Lines 477-481.

11) The Authors need to get professional help in English writing and editing.

Author Response

Rev2

Comments and Suggestions for Authors

The manuscript reports the MC1R and ASIP genotypes determining coat color in 70 Sarcidano horses. The idea of this study is interesting, but the outcome demonstrated in the manuscript is quite disappointed. The manuscript contains numerous flaws and requires extensive editorial work and a major revision.

Author’s response: Thank you for your comments and suggestions; we have made several changes following your indications and we believe that the work has now improved. The manuscript has been fully spell-checked by a native English speaker, hoping that the corrections done improved paper reading

Rev2#point 1:

1) The Authors studied 70 horses of 100 Sarcidano horses. From their description, I understood that this horse population is highly inbred, and all these horses should be considered as the relatives. Strictly speaking, the Authors determined the frequencies of the alleles and genotypes in a large pedigree rather the frequencies in a population with randomly mating animals. In that case, the study has no sense without demonstrating the pedigree of these animals determined by horse managers or microsatellite profiling.

Author’s response#point1: The microsatellite profiling is an interesting topic for future research that we are planning to pursue with the aim of safeguarding this interesting genetic resource. In the present study the purpose was to give a classification of the coat colors on a genetic basis, to better define the individual phenotypic information. It could be useful for improving the individual information in the stud book, with a view to a better knowledge of the breed. These horses do not live all together in a single group, but are distributed in different territories, manged by a few passionate breeders who are aware of the dangers of inbreeding and prevent them avoiding that a single male remains for too long within the same group. These breeders know the kinship within the individual groups at least up to the second generation, but since there is no obligation to record this information, often there is no certain information. Some Mother/Son-Daughter binomas were identified by the young age of the under-mother foals during capture discussed in the present manuscript, and they were now reported on the individual form kept by breeder.

Rev2#point 2:

2) In situation when the pedigree information or the microsatellite profiling is unavailable, the Authors need at least to provide a supplement with the individual information for each animal including gender, age, phenotypic coat color, MC1R and ASIP alleles, MC1R and ASIP genotypes, the genetic coat color.

Author’s response#point2: this set of information has been added in an Excel Supplementary file (S1).

Rev2#point 3:

3) The restriction analysis was not confirmed by the DNA sequencing of the MC1R and ASIP alleles despite the Authors mentioned that method on page 4. All the alleles have to be confirmed and the fragments of the sequencing graphs with particular SNPs or a deletion must be included.

Author’s response#point3: the attribution of the genotype to each subject was performed through the digestion or amplification of the affected fragments. The sequencing was carried out for our internal confirmation only on a few samples, by a commercial service. However, the sequencing of the fragment of the ASIP gene, (being 91 or 102 bp in length) was not useful given the large background noise which interferes in the reading, as is typical of short fragments. Therefore, we did not consider to include sequencing graphs among the figures in the manuscript. All this missing information, have now been added in the Results section. Due to our technical problem we do not currently have access to the aforementioned sequences, and we are waiting for the external lab that performed the sequencing to make them available to us again. If you still consider it useful we will integrate them as soon as they are available.

Rev2#point 4:

4) Figures 2 and 3 are unreadable and must be substantially improved. Do not include Figure 1 in the manuscript like you want to provide evidence that your PCRs work. Please, add one lane of the undigested PCR product for the comparison with all combinations of the allelic fragments in the digested PCR samples and the PCR fragments with a deletion polymorphism. Provide the better separation of the alleles on gels. It is easy to prepare new gels and take pictures of good quality.

Author’s response#point4: following your advice we have replaced the previous figures with a single figure of better quality, containing all the allele combinations and the undigested fragment.

Rev2#point 5:

5) Why did the Authors study just 70 horses? If 100 horses were available, it would be great to see the results for all animals. The Authors did not explain why 30 animals were neglected.

Author’s response#point5: This study was born several years after that in which the animals were captured. As explained in the manuscript, being a semi-feral population, capture and withdrawal are not easy activities that can be organized in a short time, but on the contrary they require a long planning and preparation. In this particular case, the opportunity to move some animals following the changed environmental conditions, led to their capture. Captures took place on different days, have affected several veterinarians who worked in different and distant capture paddocks. The individual events that prevented the blood sample from each of the captured animals are not information in our possession. Our laboratory received 70 blood samples appointed with unique code from which DNA was extracted. Only later we obtained also phenotypic information, which concerned more animals we received.

Rev2#point 6:

6) Discussion. Please, do not discuss the topics you did not analyze in your project. You did not study how both genes MC1R and ASIP work. There is no sense to provide the textbook description on this topic in Lines 333 – 389. Instead, please discuss the distribution of the MC1R and ASIP alleles in other horse breeds and wild/feral populations. Also, describe better the artificial selection of coat color in the process of horse domestication.

Author’s response#point6: The Discussion section has been modified according to your suggestion, adding more details on the coat color changes in horse domestication, and on the distribution of the MC1R and ASIP alleles in other small local population, when available.  

Rev2#point 7:

7) Many statements in the manuscript taken from other publications missed the references. For instance, the references are missed in Lines 55, 67, 261, 269, 346, 389.

Author’s response#point7: The missing References have been added, and the Reference list has been modified accordingly.

Please note that point (8) was missing in the Revision note

9) Line 112: “After a severe bushfire…”. Please, provide more details about this bushfire and how it affected the Sarcidano horse population.

Author’s response#point9: summer fires are a rather common occurrence in Sardinia. Due to the Mistral wind that blows frequently, these fires often cover very large areas with very serious damage to livestock, grazing and the environment in general, also with the escape or death of small wild species and often causing even human victims. In this particular case discussed in the present study, it was necessary to move some groups of horses in safer pastures, waiting for the areas affected by the fire to revive.

Rev2#point 10:

10) References. The Authors need to follow the journal’s requirements for the references. Please, compare the references 1 and 2 in Lines 477-481.

Author’s response#point10: References have been corrected.

Rev2#point 11:

11) The Authors need to get professional help in English writing and editing.

Author’s response#point11: The manuscript has been spell-checked and corrected by a native English speaker who offers a review service in our Department. The corrections made have been inserted in the text in track change mode and not listed in this revision note.

Reviewer 3 Report

This is an interesting study, but more relevant to breed conservation than to coat color genetics. It extends coat color documentation to a rare breed that is the target of conservation efforts.

Author Response

Rev3

Comments and Suggestions for Authors

This is an interesting study, but more relevant to breed conservation than to coat color genetics. It extends coat color documentation to a rare breed that is the target of conservation efforts.

Author’s response#: thank you for your comment, we hope this topic could be of interest for readers and scientific community.